

# Determination of the reference genes for qRT-PCR normalization and expression levels of *MAT* genes under various conditions in *Ulocladium*

Li-Guo Ma[1] and Yun Geng[2]

[1] Shandong Key Laboratory of Plant Virology, Institute of Plant Protection, Shandong Academy of Agricultural Sciences, Jinan, China
[2] Biotechnology Research Center, Shandong Academy of Agricultural Sciences, Jinan, China

## ABSTRACT

The genus *Ulocladium* is thought to be strictly asexual. One of the possible reasons for the lack of sexuality in *Ulocladium* species is the absence of the stimulus of environmental factors. Sexual reproduction in ascomycetes is controlled by a specific region in the genome referred to as mating-type locus (*MAT*) that consists of two dissimilar DNA sequences in the mating partners, termed *MAT1-1* and *MAT1-2* idiomorphs. To identify the response of *MAT* loci to environmental conditions, the mRNA transcription level of *MAT1-1-1* and *MAT1-2-1* genes was tested using qRT-PCR under different temperatures ($-20\ °C$, $-10\ °C$, $0\ °C$, $10\ °C$, $20\ °C$, $30\ °C$ and $40\ °C$), culture medias (CM, OA, HAY, PCA, PDA and V8), photoperiods (24 h light, 24 h dark, 12 h light/12 h dark, 10 h light/14 h dark and 8 h light/16 h dark), and $CO_2$ concentrations (0.03%, 0.5%, 1%, 5%, 10%, 15% and 20%). For obtaining reliable results from qRT-PCR, the most stable internal control gene and optimal number of reference genes for normalization were determined under different treatments. The results showed that there is no universal internal control gene that is expressed at a constant level under different experimental treatments. In comparison to various incubation conditions, the relative expression levels of both *MAT* genes were significantly increased when fungal mycelia were grown on HAY culture media at $0-10\ °C$ with a light/dark cycle, indicating that temperature, culture media, and light might be the key environmental factors for regulating the sexuality in *Ulocladium*. Moreover, *MAT1-1-1* and *MAT1-2-1* genes showed similar expression patterns under different treatments, suggesting that the two *MAT* genes might play an equally important role in the sexual evolutionary process.

Corresponding author
Yun Geng, gengyun1@126.com

## INTRODUCTION

It has been estimated that there are at least 1.5 million fungi species worldwide, which is six times higher than plants (*Hawksworth, 2001*). Hyphomycetes, a major taxon of fungi, including more than 450 genera, are widely distributed in land, sea, air, and soil, which account for more than 30% of the fungal kingdom (*Hawksworth, Sutton & Ainsworth, 1983*; *Kirk et al., 2008*; http://www.indexfungorum.org). However, no sexual stage has been

identified in most of hyphomycetes. For more than 450 genera of hyphomycetes, Hyphomycetales, Dematiaceous hyphomycetes, teleomorphs were found in only 67 genera.

*Ulocladium* (*Preuss, 1851*) is an anamorphic genus of the *Pleosporaceae* (*Dothideomycetes*, *Kirk et al., 2008*). Some species of *Ulocladium* are often found as pathogens or endophytes of living plants, and act as saprobes that play vital role in the decomposition and recycling of materials in natural ecosystems (*Zitter & Hsu, 1990*; *Vannini & Vettraino, 2000*). Teleomorphs of many closely allied genera of *Ulocladium* have been found, such as *Stemphylium* Wallr, *Alternaria* Nees, and *Nimbya* E.G. Simmons. However, *Ulocladium* species are thought to be strictly asexual.

Sexual reproduction in ascomycetes is controlled by mating-type locus (*MAT*) (*Coppin et al., 1997*; *Turgeon, 1998*) that consists of two idiomorphs, termed *MAT1-1* and *MAT1-2*, with identical flanking regions and dissimilar DNA sequences (*Turgeon et al., 1993*; *Cozijnsen & Howlett, 2003*). *MAT1-1-1* and *MAT1-2-1* encode proteins containing an alpha domain and a HMG (high mobility group) domain, respectively (*Turgeon, 1998*). *MAT* genes have been characterized in many filamentous ascomycetes (*Inderbitzin et al., 2005*; *Groenewald et al., 2006*; *Santos, Correia & Phillips, 2010*; *Bolton et al., 2012*), and the *MAT1-1-1* and *MAT1-2-1* genes have been obtained from 26 *Ulocladium* species (*Geng et al., 2014*). *Ulocladium* strains possess both *MAT1-1-1* and *MAT1-2-1* genes as observed in homothallic filamentous ascomycetes. However, unlike the majority of homothallic fungi, *MAT1-1* and *MAT1-2* idiomorphs are not closely linked and flanked by identical sequences in *Ulocladium* species. It has been reported that *U. botrytis MAT* genes have the ability to induce sexual recombination in *Cochliobolus heterostrophus* (*Wang et al., 2017*), indicating that *MAT* locus might be functional in *Ulocladium*, and the expression levels of *MAT* genes might directly affect sexual development in fungi. Thus, the *Ulocladium* species has the potential for sexual reproduction. However, as yet, no sexual stage has been identified in the whole genus of *Ulocladium*. One of the possible reasons for the lack of sexuality in *Ulocladium* species is the absence of environmental stimuli, including medium, light, temperature, atmospheric gases, and others, which could influence the sexual development.

Environmental factors have great influences on sexual development of filamentous fungi, as ascospore production requires special conditions, and different ascomycete isolates show variations in response to identical incubation conditions. For instance, teleomorphs of many putatively asexual fungi have been found under controlled conditions. Perithecium formation was favored at 20 °C with a 16-h daily photoperiod in *Mycosphaerella pinodes* (*Roger & Tivoli, 1996*). For *Pyrenophora tritici-repentis*, a large number of asci and ascospores were produced by incubation on senescent leaves in continuous darkness for 12 days followed by a 12-h photoperiod at 15 °C (*Friesen et al., 2003*). Several isolates of *Stemphylium* could produce mature ascocarps on hay decoction agar, 20% V-8 juice agar or weak potato-carrot agar, within cycles ranging from a few months to a year or more (*Simmons, 1969*). *Gaeumannomyces graminis* var. *tritici* that is thought to be strictly asexual in nature could produce asci and ascospores adequately in the sterile culture medium with dark/light (12 h/12 h) at 15–20 °C for 72 h

(*Holden & Hornby, 1981*). The ability of sexual reproduction in *Candida albicans* was enhanced with high temperature, high $CO_2/O_2$, and darkness (*Ramírez-Zavala et al., 2008*; *Whiteway, 2009*).

Quantitative real-time reverse transcription polymerase chain reaction (qRT-PCR) is a valuable tool to quantify the transcript expression levels of a gene in the cells of different tissues under specific experimental conditions. To remove the non-biological variation caused by experimental deviations, inhibitory compounds, RNA isolation or reverse transcriptase efficiency, it is necessary to appropriate normalize the qRT-PCR data. The most common way to normalize the data generated by qRT-PCR is to use proper internal reference genes (*Zitter & Hsu, 1990*). Many housekeeping genes (HKGs), such as β-actin (*Actin*), glyceraldehyde-3-phosphate dehydrogenase (*GAPDH*), β-tubulin (*Tub-b*), and 18S ribosomal RNA were frequently used as internal controls for gene expression analysis. However, several studies have indicated that expression levels of such reference genes also vary even under the same experimental conditions (*Suzuki, Higgins & Crawford, 2000*; *Lee et al., 2002*). In general, there is no universal internal control gene that is expressed at a constant level under different experimental conditions (*Peters et al., 2007*; *Mitter et al., 2009*). Therefore, to avoid erroneous results, the validity of candidate reference genes under specific experimental conditions must be determined.

In this study, to find the most stable internal control gene (s) for normalization of qRT-PCR in *Ulocladium* under different environmental conditions, the expression of seven frequently used HKGs, including *Actin*, *β-tubulin*, *EF-1α*, *GAPDH*, *RL13*, *TBP*, and *UBC* were assessed. In addition, to identify the response of *MAT* loci to environmental conditions in *Ulocladium*, the mRNA transcription levels of *MAT1-1-1* and *MAT1-2-1* genes were analyzed.

# MATERIALS AND METHODS

## *Ulocladium* strains and growth conditions

The strains of *U. botrytis* (CBS 198.67) and *Stemphylium botryosum* (teleomorph *Pleospora tarda*) (CBS714.68) were used for test. Owing that *Ulocladium* is closely allied to the genus *Stemphylium* Wallr, the teleomorph of *Stemphylium botryosum* was chosen as sexual representative. All samples were obtained from the culture collection of the Westerdijk Institute, Utrecht, The Netherlands. For each strain, cultures were grown on potato–carrot agar (PCA; 20 g white potato boiled and filtered, 20 g carrot boiled and filtered, 20 g agar, one L distilled water) at 25 °C for 10 days.

## Treatments

For temperature treatments, small squares of fungal mycelia were placed in the middle of the petri dishes (9 cm in diameter) containing PCA medium, and the cultures were grown at 25 °C for 10 days. Cultures were then exposed to seven different temperature treatments: −20 °C, −10 °C, 0 °C, 10 °C, 20 °C, 30 °C, and 40 °C, with a daily photoperiodic cycle of 14 h light and 10 h dark for 7 days. Fungal mycelia were scraped from the surface of the agar firstly, and then stored at −80 °C until RNA extraction. Three biological replicates were used for each condition.

For medium treatments, small squares of fungal mycelia were placed in the middle of the petri dishes (9 cm in diameter) containing PCA, PDA (200 g white potato, 20 g glucose, 20 g agar, one L distilled water), CMA (200 g corn meal, 20 g agar, one L distilled water), HAY (10 g hay, 20 g agar, one L distilled water), OA (100 g oatmeal, 20 g agar, one L distilled water) and V8 (150 ml V8 juice, 1.6 g $CaCO_3$, 20 g agar, one L distilled water) medium separately, and the cultures were grown at 10 °C in a 12:12 light dark regime for 2 weeks. Fungal mycelia were scraped from the surface of the agar firstly, and then stored at −80 °C until RNA extraction. Three biological replicates were used for each condition.

For photoperiod treatments, small squares of fungal mycelia were placed in the middle of the petri dishes (9 cm in diameter) containing HAY, and then exposed to different photoperiods, respectively: 24 h light, 24 h dark, 12 h light: 12 h dark, 10 h light: 14 h dark, 8 h light: 16 h dark, at 10 °C for 2 weeks. Fungal mycelia were firstly scraped from the surface of the agar, and then stored at −80 °C until RNA extraction. Three biological replicates were used for each condition.

For $CO_2$ treatments, small squares of fungal mycelia were placed in the middle of the petri dishes (9 cm in diameter) containing HAY, and the cultures were grown at 25 °C for 10 days. Cultures were then respectively exposed to seven $CO_2$ concentrations: 0.03%, 0.5%, 1%, 5%, 10%, 15%, and 20%, at 10 °C with a light/dark cycle for 7 days. Fungal mycelia were firstly scraped from the surface of the agar, and then stored at −80 °C until RNA extraction. Three biological replicates were used for each condition.

## Total RNA extraction and reverse transcription

Mycelia of *U. botrytis* and ascomata of *S. botryosum* were collected for total RNA extraction. Total RNA was extracted with Trizol reagent (Invitrogen, Carlsbad, CA, USA) according to the manufacturer's instructions, followed by RNase-free DNase treatment (Takara, Tokyo, Japan). The RNA concentrations were quantified by a spectrophotometer (spectra-Max plus 384), and the RNA integrity was assessed by 1% agarose gel electrophoresis. RNA samples were used as template for PCR amplifications to confirm that no genomic DNA contamination appeared in the samples using primers 5′-CCACCATCCACTCTT ACACCG-3′ and 5′-TGACCTTGCCGACAGCCT-3′. The SuperScript First-Strand Synthesis System (Invitrogen, Carlsbad, CA, USA) for RT-PCR is optimized to synthesize first-strand of cDNA use with 3 μg of total RNA according to manufacturer's instructions. The obtained cDNA samples were then diluted at 1:5 in nuclease-free water. The reverse transcriptions were performed in triplicate for all treatment conditions.

## qRT-PCR

qRT-PCR was performed using ICycler IQ real-time PCR detection system (Bio-Rad, Hercules, CA, USA) and SYBR primer Script RT-PCR kit (TakaRa, Tokyo, Japan) with the following cycling parameters: an initial denaturation step at 95 °C 10 min, 45 cycles of 10 s at 95 °C, 55 s at 60 °C, and finally 45 s at 72 °C. For qRT-PCR reaction mixtures contained 12.5 μl of the 2 × SYBR Green PCR master mix, 800 nM of each primer and 2.5 μl of cDNA template in a total volume of 25 μl. The primers were designed using Primer Express 3.0 for qRT-PCR. Three replicates were performed for all reactions, and

negative controls were included for each gene. Threshold cycle (CT) values were exported to Microsoft Excel, and the fold changes of each gene were calculated using $2^{-\Delta\Delta Ct}$ equation.

### Determining expression stability of reference genes

Average gene expression stability (M) and pairwise variation (V) were calculated by geNorm (version 3.5) program. Genes with the lowest M values have the most stable expression (*Vandesompele et al., 2002*). Furthermore, normalization factors ($NF_n$) and $V_n/V_{n+1}$ value were calculated to estimate the optimal number of controls used in each single experiment.

### Statistical analysis

Statistical analyses were performed by analysis of variance (ANOVA) using SPSS v. 13.0 (SPSS, United States).

## RESULTS

### Selection of HKGs

Although morphological and phylogenetic analysis of *Ulocladium* has been studied well, little effort has been made for gene expression studies using qRT-PCR. Seven most commonly used HKGs in other ascomycetes were selected as candidate reference genes in *Ulocladium*: β-actin (*Actin*), β-tubulin (*Tub-b*), translation elongation factor 1α (*EF-1α*), glyceraldehyde-3-phosphate dehydrogenase (*GAPDH*), 60S ribosomal protein L13 (*RL13*), TATA box binding protein (*TBP*), and ubiquitin-conjugating enzyme (*UBC*). All the primer sets were confirmed by the appearance of a single, dominant peak in the qRT-PCR dissociation curve analyses (Fig. S1), indicating the specificity of the qRT-PCR reactions.

### Ranking the expression stability of reference genes under different treatments

The geNorm software was then used to compare the transcription levels of the selected seven candidate reference genes and identify the most stable gene under different experimental conditions. The average expression stability (M) values of the selected genes were ranked. The results showed that there was no universal internal control gene that is expressed at a constant level under different experimental treatments.

For temperature treatments, all candidate reference genes that reached high expression stability showed relative low M values (<1). *EF-1α* and *Actin* were the most stable genes, while *RL13* was the most unstable gene. The ranking of the expression stability of selected genes in different temperatures as follows: *EF-1α, Actin>GAPDH>UBC>TBP>*β-*tubulin> RL13* (Fig. 1A).

For medium treatments, the *M* values for all investigated genes were lower than 1.5. *TBP* and *EF-1α* were ranked as the most stable reference genes and *Actin* was the most unstable gene. The ranking of the expression stability for different medium treatments as follows: *TBP, EF-1α>GAPDH>* β-*tubulin>UBC> RL13> Actin* (Fig. 1B).

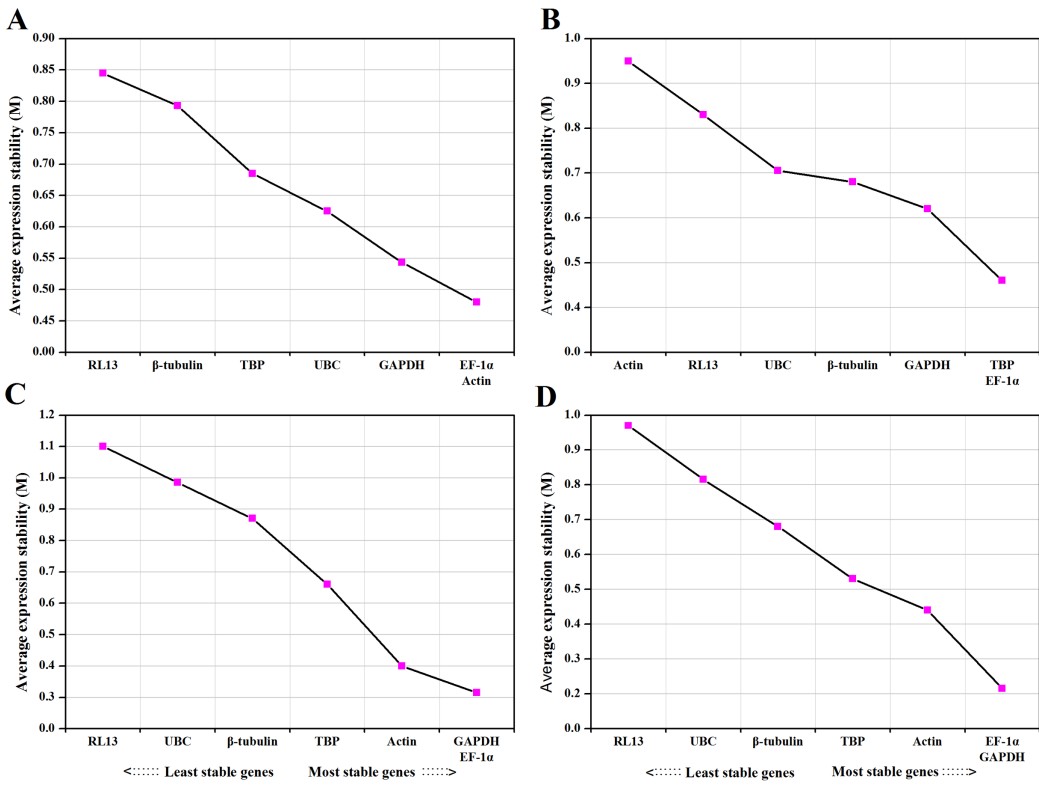

**Figure 1 Average expression stability values (M) and ranking of control genes under different temperature (A), medium (B), photoperiod (C) and $CO_2$ (D) treatments.** A lower *M* value indicates more stable expression.

For photoperiod treatments, the *M* values for all supposed reference genes were lower than 1.5. The most stable genes were *GAPDH* and *EF-1α*. The ranking of the expression stability for different photoperiod treatments as follows: *GAPDH, EF-1α>Actin>TBP>β-tubulin>UBC> RL13* (Fig. 1C).

For $CO_2$ treatments, the *M* values for all investigated genes were lower than 1.5. *EF-1α* and *GAPDH* were the best-ranked genes, and *RL13* was the worst scoring gene. The ranking of the expression stability for different $CO_2$ treatments as follows: *EF-1α, GAPDH>Actin>TBP> β-tubulin>UBC>RL13* (Fig. 1D).

To estimate the optimal number of reference genes for normalization under different conditions, normalization factors ($NF_n$) and $V_n/V_{n+1}$ value were calculated. The default cut-off threshold is 0.15 (*Vandesompele et al., 2002*), indicating it is unnecessary to select an additional reference gene when $V_n/V_{n+1}$ below 0.15. As shown in Fig. 2, two control genes for "photoperiods" (Fig. 2C), three reference genes for "$CO_2$ concentrations" (Fig. 2D), and the four most stably expressed genes for other treatments (Figs. 2A and 2B) were required for reliable normalization of qRT-PCR data.

## Transcriptional levels of the *MAT* genes under different treatments

For temperature treatments, the transcription levels of *MAT1-1-1* and *MAT1-2-1* genes were investigated at 10 °C intervals from −20 to 40 °C. The transcription levels of *MAT*

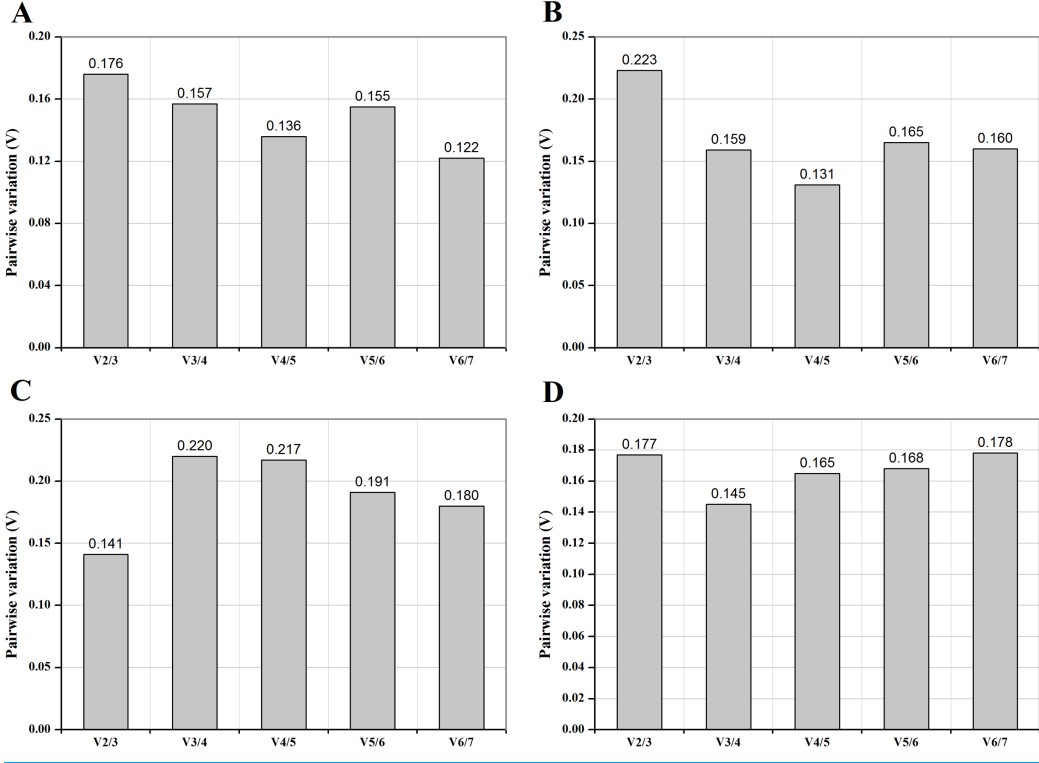

**Figure 2 Pairwise variation (V) of control genes under different temperature (A), medium (B), photoperiod (C) and CO$_2$ (D) treatments.** The pairwise variation ($V_n/V_{n+1}$) was analyzed between the normalization factors NF$_n$ and NF$_{n+1}$ to determine the optimal number of control genes for normalization.

genes gradually increased from −20 to 0 °C, and then eventually reached a peak at 10 °C (Fig. 3). No significant differences were observed between 0 °C and 10 °C. These results showed that the expression of *MAT* genes was induced at 0–10 °C.

For medium treatments, the *MAT1-1-1* and *MAT1-2-1* transcription levels of mycelia for *Ulocladium* on different kinds of culture medias (CM, OA, HAY, PCA, PDA, V8) were tested. *MAT* genes showed the highest transcription levels on HAY media, and did not show obvious difference among the other culture medias (Fig. 4). The nutritional value of the HAY is the lowest relative to the remaining media, suggesting that the nutrient-poor culture conditions could cuase a significant increase in the expression of *MAT* genes.

For photoperiod treatments, the transcription levels of *MAT1-1-1* and *MAT1-2-1* genes were investigated when were exposed to different photoperiods (24 h light, 24 h dark, 12 h light:12 h dark, 10 h light:14 h dark, 8 h light:16 h dark). The expression of *MAT* genes with a light/dark cycle was obviously higher than those of in other photoperiod treatments (Fig. 5). These results indicated that *MAT* loci could be activated by a light/dark cycle.

For CO$_2$ treatments, the transcription levels of *MAT1-1-1* and *MAT1-2-1* genes were tested when were exposed to different CO$_2$ concentrations (0.03%, 0.5%, 1%, 5%, 10%, 15%, and 20%). The transcription levels of *MAT* genes remain unchanged after many repetitions (Fig. 6), indicating that the *MAT* expression could not be triggered by CO$_2$.

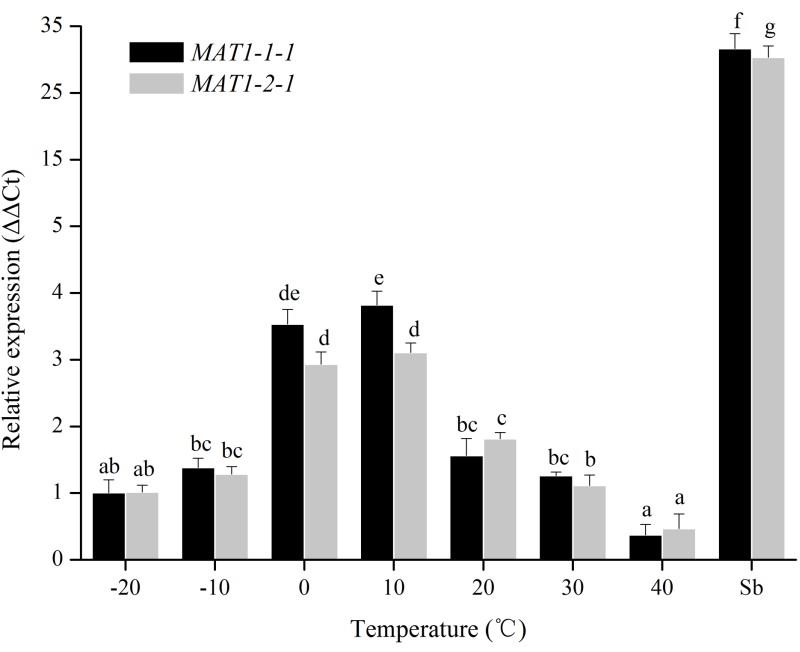

**Figure 3 Expression levels of the *MAT1-1-1* and *MAT1-2-1* genes under different temperatures.**
Sb: Expression levels of the *MAT1-1-1* and *MAT1-2-1* genes in *Stemphylium botryosum*. Different letters indicate significant differences ($P < 0.05$). Error bars represent standard errors.

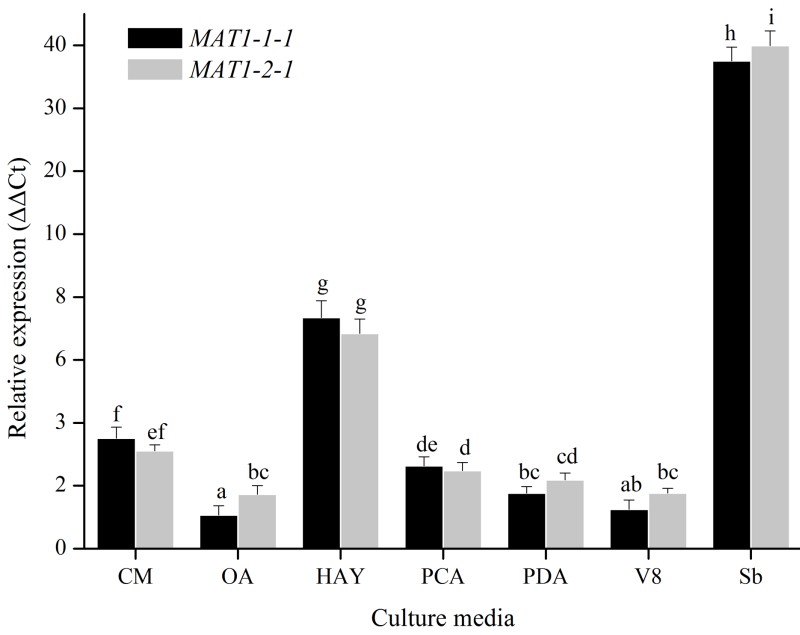

**Figure 4 Expression levels of the *MAT1-1-1* and *MAT1-2-1* genes on different culture medias (CM, OA, HAY, PCA, PDA and V8).** Sb: Expression levels of the *MAT1-1-1* and *MAT1-2-1* genes in *Stemphylium botryosum*. Different letters indicate significant differences ($P < 0.05$). Error bars represent standard errors.

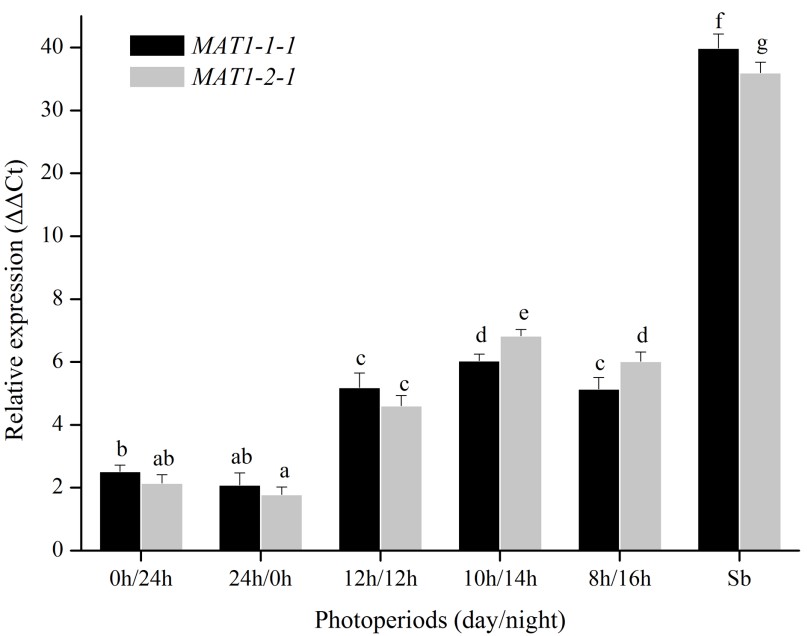

**Figure 5 Expression levels of the *MAT1-1-1* and *MAT1-2-1* genes under different photoperiods (24 h light, 24h dark, 12 h light/12 h dark, 10 h light/14 h dark and 8 h light/16 h dark).** Sb: Expression levels of the *MAT1-1-1* and *MAT1-2-1* genes in *Stemphylium botryosum*. Different letters indicate significant differences ($P < 0.05$). Error bars represent standard errors.

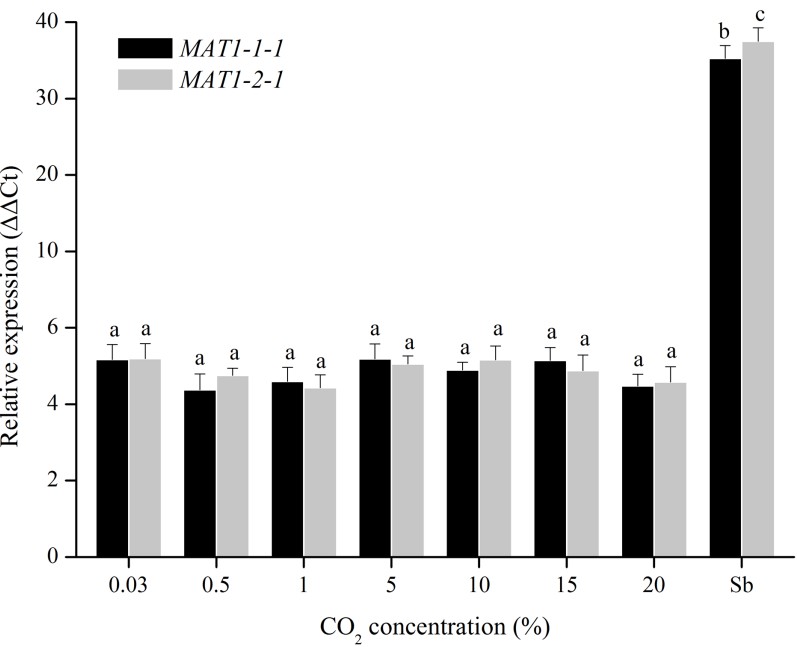

**Figure 6 Expression levels of the *MAT1-1-1* and *MAT1-2-1* genes under different $CO_2$ concentrations (0.03%, 0.5%, 1%, 5%, 10%, 15% and 20%).** Sb: Expression levels of the *MAT1-1-1* and *MAT1-2-1* genes in *Stemphylium botryosum*. Different letters indicate significant differences ($P < 0.05$). Error bars represent standard errors.

Overall, the relative expression levels of both *MAT* genes were significantly increased on HAY culture media at 0–10 °C with a light/dark cycle, indicating that the temperature, culture media, and light are the key environmental factors that affect the sexual development of *Ulocladium*. Similar expression patterns for *MAT1-1-1* and *MAT1-2-1* genes were found, suggesting that the two *MAT* genes might play an equally important role in the sexual process. Moreover, the expression levels of *Stemphylium botryosum MAT* genes are significantly higher than those of *U. botrytis*, indicating that environmental factors alone are insufficient to induce sexual progeny for *Ulocladium*.

## DISCUSSION

The qRT-PCR is now increasingly becoming the method of choice for accurate and sensitive quantification of transcript expression levels. For valid qRT-PCR analysis, selection of appropriate internal control genes for accurate normalization is required. However, the expression of reference genes was instable under different experimental conditions, and the expression patterns of the same reference varied in different fungal species. Thus, it is necessary to determine the validity of candidate reference genes before normalization. Moreover, quantitative analysis of gene expression should be related to several housekeeping genes in parallel (*Schmid et al., 2003*), thus it is important to determine the optimal number of internal control genes required for accurate data normalization.

*Actin*, *Tub-b*, *EF-1α*, *GAPDH*, *RL13*, *TBP* and *UBC* were found as most reliable HKGs for normalization of qRT-PCR data in many ascomycetes. However, it remained unclear whether these traditionally used reference genes are in fact the best possible choices for normalizing gene expression data in *Ulocladium*. To get reliable results from qRT-PCR, the most stable internal control genes, as well as the optimal number of reference genes for normalization were determined under temperature, medium, photoperiod, and $CO_2$ treatments in *Ulocladium*. Consistent with previous studies, our results showed that there was no universal internal control gene that is expressed at a constant level under different experimental treatments (*Suzuki, Higgins & Crawford, 2000*; *Lee et al., 2002*; *Boxus, Letellier & Kerkhofs, 2005*).

Sexual reproduction could enhance the adaptability to environmental stress, which plays a prominent role in the evolution and continuation of species (*Ashton & Dyer, 2016*). Sexual recombination is dominant in harsh habitats, including desiccation, high temperatures, fungicides and so on (*Bowden & Leslie, 1999*; *Dyer & O'Gorman, 2012*). Asexual lineages were thought to be evolutionary dead ends (*Wik, Karlsson & Johannesson, 2008*), though they could complete population spread in a short time. However, the sexual stage of *Ulolcladium* has not been found yet. One of the possible reasons for the absence of sex in *Ulocladium* species is the lack of suitable environmental conditions influencing sexual development.

Environmental factors play an important role in regulating sexual differentiation and development of fungi. *Hawker (1966)* systematically discussed the effects of environmental conditions on the sexual reproduction of filamentous ascomycetes, and proposed that the environment suitable for mycelia development or asexual reproduction are not

necessarily adapt to sexual cycle and the later development of teleomorphs. In recent years, mycologists have discovered the teleomorphs in several filamentous ascomycete species, which were thought to be strictly asexual from Antarctica, Arctic, snow-covered mountains, nutrient-poor salt lakes, deserts and plateau areas. Moreover, a few putatively asexual species have been successfully induced teleomorphs in artificial stress conditions (poor nutrition, low temperature, strong ultraviolet light, and high $CO_2/O_2$) (*Simmons & Roberts, 1993*; *Câmara, O'Neill & Van Berkum, 2002*). Environmental stresses may serve as a trigger to induce the expression of *MAT* genes, and the activated mating signaling could then promote the sexual evolution and development of asexual hyphomycetes.

Temperature, nutritional environments, photoperiod and $CO_2$ concentrations have prominent effect on the sexual recombination of filamentous ascomycetes (*Swart et al., 2001*; *Klich, 2002*; *Dyer & O'Gorman, 2012*). It has been found that 10–15 °C was favorable for the formation of apothecium in *Monilinia* (*Casals et al., 2010*), and the optimum temperature for sexual reproduction of *Aspergillus fumigatus* was 30 °C (*O'Gorman, Fuller & Dyer, 2008*). Low concentrations of carbon or nitrogen can promote sexual reproduction of filamentous ascomycetes (*Han et al., 1994*, *2003*). *Gelasinospora reticulospora* and *Alternaria* species could both produce a large number of perithecia under the light or dark conditions (*Simmons & Roberts, 1993*; *Chamberlain & Ingram, 1997*). In addition, on solid media, the number of asci improved significantly with increasing $CO_2$ concentration in *Aspergillus nidulans* (*Zonneveld, 1988*). For *Ulocladium* species, extensive mating experiments were performed to induce sexual recombination under different temperatures, culture medias, photoperiods, and $CO_2$ concentrations. No reproductive structures were observed despite the long incubation period of more than 6 months.

Though the teleomorph formation has never been successful, we found that the transcription levels of *Ulocladium MAT1-1-1* and *MAT1-2-1* genes were up-regulated on HAY culture media at 0–10 °C with a light/dark cycle. *MAT* genes that are the key regulators for sexuality in filamentous ascomycetes are functional in *Ulocladium* (*Geng et al., 2014*; *Wang et al., 2017*), which could be activated by several environmental cues, and then control the sexual development in fungi. In addition, though specific environmental conditions can promote the expression of *MAT* genes of this genus, compared to *Stemphylium botryosum*, the significantly lower levels suggesting that environmental factors alone are insufficient to induce sexual progeny. Further studies should be taken to reveal the key regulatory networks related to fungal reproduction by multi-omics analysis, and clarify the role of environmental factors in sexual development.

## CONCLUSIONS

In this study, the most stable internal control gene and optimal number of reference genes for normalization were determined under temperature, medium, photoperiod, and $CO_2$ treatments in *Ulocladium*. In comparison to various incubation conditions, the relative expression levels of both *MAT* genes were significantly increased when fungal mycelia were grown on HAY culture media at 0–10 °C with a light/dark cycle, indicating that temperature, culture media, and light might be the key environmental factors that affect the sexual development of *Ulocladium*.

## ACKNOWLEDGEMENTS

We thank the herbarium of the Westerdijk Institute for providing some valuable isolates in this study. We express gratitude to Dr. Xiuguo Zhang (Shandong Agricultural University) for his valuable comments and suggestions.

### Funding

This work was supported by Young Talents Training Program of Shandong Academy of Agricultural Sciences (CXGC2018E04), National Key R&D Program of China (2016YFD0300700, 2017YFD0201700), National Natural Science Foundation of China (31400019) and Natural Science Foundation of Shandong Province (BS2015SW020). The funders had no role in study design, data collection and analysis, decision to publish, or preparation of the manuscript.

### Grant Disclosures

The following grant information was disclosed by the authors:
Shandong Academy of Agricultural Sciences: CXGC2018E04.
National Key R&D Program of China: 2016YFD0300700, 2017YFD0201700.
National Natural Science Foundation of China: 31400019.
Natural Science Foundation of Shandong Province: BS2015SW020.

### Competing Interests

The authors declare that they have no competing interests.

### Author Contributions

- Li-Guo Ma performed the experiments, analyzed the data, authored or reviewed drafts of the paper, and approved the final draft.
- Yun Geng conceived and designed the experiments, analyzed the data, prepared figures and/or tables, and approved the final draft.

### Data Availability

    The raw measurements are available in the Supplemental Files.

### Supplemental Information

Supplemental information for this article can be found online at http://dx.doi.org/10.7717/peerj.10379#supplemental-information.

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
