# Peer review of "Determination of the reference genes for qRT-PCR normalization and expression levels of MAT genes under various conditions in Ulocladium"

_PeerJ, doi:10.7717/peerj.10379_

## Round 0.1 · original submission · Major Revisions

It is important you address reviewer 2's comments and get assistance with English. Reviewer 1 gave you detailed information.

Reviewer 1 ·

Basic reporting

1. The English in this manuscript needs to be greatly improved for overall readability and comprehension.
• Some examples where the language could be improved include (but are not limited to) sentences on lines 41-43, 205, 209, 217, 267-269, 284-286, and 315.

2. The introduction could use some improvement.
• In many places, more recent literature can be cited. Some examples are:
i. Lines 85: Most recent literature cited was from 2006
ii. Lines 89 & 90: Most recent literature cited was from 1998 & 2004
iii. Line 90: No reference provided for “… and fungi.”.
iv. Line 329: No reference provide for “… functional in Ulocladium.”

• The introduction needs to better explain the sexual strategy used by these species. For example, it is not immediately clear that the Ulocladium species being studied exhibit primary homothallism and harbour both MAT1-1-1 and MAT1-2-1 in a single isolate.

• The reference list needs to be checked and corrected in many places. Some examples are:
i. Line 388: “Insights” should be capitalized
ii. Line 414: “The” should be capitalized
iii. Lines 455 – 456: “Diaporthe”, “Phomopsis” and “in vitro” should be italicized, “Their” should be capitalized
iv. Line 493: “mat” and “Neurospora” need to be italicized
v. Line 503: “Aspergillus” needs to be italicized

3. The structure does confirm to PeerJ standards as well as discipline norms.

4. The figure are relevant and illustrate the results. However, the titles and descriptions of the figures are insufficient.
• Figure 1: “Environmental concentrations”  “Environmental conditions”.
• Figure 1: In each graph, the title is “Average expression stability value of remaining control genes”. In this context, what does “remaining” mean, as all 7 genes are included in each condition? Furthermore, these titles can be removed as the overall figure has a relevant title.
• Figure 1: The y-axis labels should include “M” in brackets.

• Figure 2: Each graph has its own title (Determination of the optimal number of control genes for normalization) which can be removed as the overall figure has a relevant title.
• Figure 2: The y-axes need labels and the x-axis labels are not descriptive.

• Figure 3: The y-axis needs a unit of measurement (ΔΔCt)
• Figure 3: Correct “Expression levels of the MAT1-1-1 and MAT1-2-1 genes in Pleospora tarda”.
• Figure 3: What is the relevance of “CK” as the abbreviation for P. tarda?
• Figure 3: Although error bars are shown, statistically significant differences should also be indicated on the graphs- ie: if there is a significant difference in MAT1-1-1 expression between -20'C and 0'C, this should be indicated using either asterisks (*) or symbols (a, b).

• Figure 4: All of the same issues as with Figure 3
• Figure 4: The x-axis label is incorrectly spelt: “Clture”  “Culture”

• Figure 5: All of the same issues as with Figure 3
• Figure 5: Correct “Expression levels of MAT1-1-1 and MAT1-2-1 genes under different photoperiods”.

• Figure 6: All of the same issues as with Figure 3
• Figure 6: Correct “Expression levels of MAT1-1-1 and MAT1-2-1 genes under different CO2 concentrations”.

5. Raw data was supplied

Experimental design

1. The research presented in this manuscript is original and primary research that represents an impressive addition to the work conducted on the MAT genes and sexual cycle of Ulocladium species in Geng et al 2014 and Wang et al 2017.

2. The research question(s), while relevant and meaningful, need to be better defined. I suggest rewriting the final paragraph of the introduction to better describe what the study aimed to do. It was unclear whether the main aim was to identify usable reference genes or to determine the expression of the MAT genes under different environmental conditions. It is certainly clear that this study aims to address a particular knowledge gap and this was more clearly explained.

3. The research was performed to a high technical standard.

4. Overall, the methods are described with sufficient detail. However, the methods could improve upon in the following ways:
• Line 116 & 256: Two different Ulocladium species are used- however, it is not possible to determine which species is represented in the results. In the discussion, it is further stated that “different fungal species” showed different gene expression levels but this data is not presented.
• Line 117: Explain the relevance of using Pleospora tarda as a sexual representative.
• Under what conditions was P. tarda grown to get the MAT gene expression values?
• Lines 121 – 122: The final sentence of the first paragraph of the materials and methods seems misplaced as RNA is extracted after each of the environmental conditions.
• Line 125: Petri dishes (9cm)- is this the plate diameter?
• Please explain why the treatment experiments were conducted for different time periods: temperature (7 days- Line 127), medium and photoperiod (14 days- Lines 136 & 142), and CO2 (10 days- Line 146).
• What does “proper temperature” (Lines 135, 141 & 148), “optimum media” (Lines 140 & 146) and “proper photoperiod” (Line 148) mean in this context?
• Line 152: What exact material from P. tarda was used to extract RNA? The asci? Or the ascomata? (see Major Flaw comments below)
• Lines 158 – 159: Please add more information regarding what PCRs were done to confirm that no gDNA contamination was present.
• Lines 172 – 174: Move the sentence “Melting curve … reactions” to the results section
• Line 176: Correct the equation to read 2-ΔΔCt and not “2- ΔΔC<SUB>t</SUB>”.
• What statistical methods did you use to compare the expression values across experimental treatments? Ie: how did you come to the conclusion that, for example, MAT genes are expressed at a statistically significant higher level on HAY media than on PCA media?
• Line 182: Add a sentence better describing the expression stability measure (M)- what is considered a low/high value?

Validity of the findings

1. All the underlying data was provided.

2. The majority of the concluding points, including the importance of temperature, media type, photoperiod and CO2 concentrations to the sexuality of Ulocladium species, should be identified as speculative. Additional, the authors should include a statement on what further studies can be done to supporting these speculative conclusions.

3. Many of the conclusions are supporting the results.
• See below the “Major flaw” section in the general comments for author

Additional comments

The manuscript entitled “Low expression levels of MAT genes may be one of the major causes of asexuality in Ulocladium” is a primary research article that makes two important contributions to the field:
1) The manuscript provides important and novel knowledge regarding the use of various housekeeping genes as reference genes for RT-qPCR in Ulocladium species.
2) Additionally, the manuscript sought to identify important environmental conditions that may be conducive to sexual development in Ulocladium species.

Below, I have detailed some suggestions and changes I think are necessary to improve the quality of the manuscript.

Major Flaw:
• One of the major flaws of this manuscript is that the authors have strongly linked high expression levels of the MAT genes to the potential for sexual reproduction. Unless they have reference to support this theory, I suggest that, in fact, MAT gene expression is fairly low in vegetative mycelia and increases upon the onset of sexual development. Thus, I do not think that their claim that “low level expression of the MAT genes is a major cause of asexuality” stands true.

• One of the “supporting results” for this low level expression is the comparison of MAT gene expression in the two Ulocladium species to the expression of the MAT genes in P. tarda. However, from what I can gather from the methods section, RNA extracted from P. tarda was extracted from sexual tissue- which would explain the high levels of MAT gene expression. Thus, this comparison is not biologically relevant.

• Similarly, the authors go as far to use MAT gene expression as a proxy for sexual reproduction. On a number of occasions (Lines 229, 244, 288, 296, 310, 319,), the authors say that conditions that showed the highest MAT gene expression are conducive to sexual reproduction. This is not true, because sexual reproduction and the production of sexual structures was not achieved during the course of this research.

• The authors thus need to tone down the relationship between MAT gene expression levels and the potential for sexual reproduction. Instead, the manuscript should focus on the important contribution of potential HKG for RT-qPCR analysis as well as the different environmental conditions that are not conducive to sexual development.

Minor Flaws:
1. In many cases, the authors use the terms anamorph and teleomorph incorrectly. These terms refer to particular morphs of an isolate- not sexualized tissue.
• Line 152: “teleomorphs of P. tarda” – do they just mean sexual tissue? Ascomata?
• Line 244: “teleomorphs are not sensitive” – teleomorphs were not seen?
• Line 288: “production of teleomorphs” – teleomorphs were not seen?

2. Similarly, the authors mention that sexual structures have been found in asexual species. This clearly means that these species are no asexual. Where this has been stated, adding phrases like “thought to be asexual” would better describe these species (Lines 56, 64, 271 and 272).

3. The authors state at various times (Lines 239 & 310) that the results indicate that light is necessary for sexual development- however, the results actually indicate that a light/dark cycle is important. MAT gene expression is highest when a light/dark cycle is used, not when only light is provided.

4. Line 154 and elsewhere: Please ensure that there is more consistency in your use of “real time”, “reverse transcriptase”, and the abbreviations “RT”, “RT-qPCR” and “qRT-PCR”. Once you have mentioned the technology and its abbreviation once (Line 81), use the abbreviation for all other mentions.

5. Lines 53 & 278: Remove the use of “etc” and replace instead with something like “and others”

6. Lines 326 – 328: The authors state that they “performed extension mating testing by confronting cultures of the same and different strains”. This is never referenced in the methods. Instead, the only methods include plating out a single isolate. This should either be expanded upon or removed.

7. Minor grammatical/spelling errors
• Line 59: Producted  produced
• Line 71: “… and MAT1-2, with”
• Line 73: “… domain, respectively”
• Line 74: lots of  many
• Line 79: “… MAT locus was functional”
• Line 96 – 97: “They are involved in basic”
• Line 100, 201 & 262: “… control gene that is expressed”
• Line 102: “… reference genes under”
• Line 119: “Utrecht, The Netherlands.”
• Line 198: “The seven candidate…”
• Line 292 – 293: “Hypomyces solani var. curcurbitae”
• Line 249: Remove “evolutionary”
• Lines 270 – 271: “… recognized as strictly”
• Line 315: “… production of asexual spores”

·

Basic reporting

The manuscript may be of interest but this cannot be verified since legends are missing and the figures are by no-way self-explanatory.

The introduction contains a long paragraph on housekeeping genes that seems to directly belong in a textbook, while 76 references occupy nearly one third of the entire document.

The authors do not comply with the one-fungus-one-name paradigm effective as of 1 Jan 2013.

Experimental design

The MS is rather methodological and the results are not very convincing. Both mat1-1-1 and mat 1-2-1 show similar expression patterns under all conditions. Moreover, the structure of the idiomorphs (homothallic vs. heterothallic structure) is missing, which makes things difficult to follow.

Validity of the findings

The authors show qualitative data on housekeeping genes that seem robust.

The expression profiles of both mating type genes does not seem to vary significantly.

Additional comments

The authors are encouraged to have their manuscript reviewed by native English speaking researchers before submission in the future.

Some remarks in the accompanying file might be helpful in improving the document.

---

## Round 0.2 · Minor Revisions

Your manuscript has met the considerations of the reviewers with the exception of a few minor points. Please revise it accordingly.

Reviewer 1 ·

Basic reporting

Lines 43 – 44: The sentence “Of all known… 76 genera” does not make sense. Please rewrite.

Line 56: The sentence should state “MAT1-1-1 and MAT1-2-1 encode proteins…” – It currently states “MAT1-1 and MAT1-2” which are the idiomorphs, not the genes.

Lines 69 – 71: Change the sentence to read “… is the absence of environmental stimuli, including medium…”

Line 78: Incubating  Incubation

Line 101: Genera  genus

Line 112 – 113: CBS is now called The Westerdijk Institute

Line 147 – 149: Please indicate which PCRs (ie: what primers were used) were performed to determine whether genomic DNA was present. This was not adequately dealt with in the first review. Simply stating that no bands appeared does not confirm the absence of gDNA if the readers don’t know what primers were used.

Experimental design

No comment.

Validity of the findings

No comment.

---

## Round 0.3 · accepted · Accept

You have been able to meet the suggestions of the reviewer.